# Efficacy of *S*-metolachlor + glyphosate for Weed Control in Different Levels of Eucalyptus Straw

Tayna Sousa Duque [1], Fernanda Santos Oliveira [1,*], Iasmim Marcella Souza [1], Bruno Caio Chaves Fernandes [2], Luma Lorena Loureiro da Silva Rodrigues [2], Daniel Valadão Silva [2] and José Barbosa dos Santos [1]

1   Departamento de Agronomia, Universidade Federal dos Vales do Jequitinhonha e Mucuri, Diamantina 39100-000, Minas Gerais, Brazil; tayna.duque@ufvjm.edu.br (T.S.D.); iasmim.marcella@ufvjm.edu.br (I.M.S.); jbarbosa@ufvjm.edu.br (J.B.d.S.)
2   Departamento de Agronomia e Ciências Vegetais, Universidade Federal Rural do Semi-Árido, Mossoró 59625-900, Rio Grande do Norte, Brazil; brunocaio@ufersa.edu.br (B.C.C.F.); luma_lorena@hotmail.com (L.L.L.d.S.R.); daniel.valadao@ufersa.edu.br (D.V.S.)
*   Correspondence: oliveira.fernanda@ufvjm.edu.br; Tel.: +55-31-9-99430-5274

**Abstract:** The occurrence of weeds in eucalyptus plantations can cause losses in productivity. Chemical control is widely used, but the efficiency of herbicides depends on management and environmental factors. This study aims to evaluate the efficiency of S-metolachlor + glyphosate in the control of grasses in different densities of eucalyptus straw and with simulated rainfall after application of the product. The experiment was conducted in a randomized block design, factorial, with four replications. The first factor represented 0; 1.06 + 0.79 e 2.12 + 1.59 kg i.a. ha$^{-1}$ of the commercial dose of *S*-metolachlor + glyphosate, the second 0; 5 and 10 tons ha$^{-1}$ of straw and, the third 25 and 50 mm of water depth applied in soil with a mix of grasses previously sowed. The evaluations carried out were fresh mass, dry mass, and visual analysis of the control percentage. The fresh and dry mass and the grasses' dry mass/water ratio decreased with increasing herbicide dosage and straw density. The treatments without straw and with the herbicide application had the highest percentages of control, the highest in the dosage of 2.12 + 1.59 kg i.a. ha$^{-1}$ of *S*-metolachlor + glyphosate. Applying different water depths (25 mm or 50 mm) did not influence the control. In conclusion, it was observed that the isolated straw promoted the control of grasses. However, in treatments that included straw and herbicide, there was a decrease in the efficiency of the product, which suggests an antagonism between the vegetation cover and *S*-metolachlor + glyphosate.

**Keywords:** forestry; grasses; herbicide; simulated rain





## 1. Introduction

The genus *Eucalyptus* stands out as the most planted forest crop in the world [1,2]. Brazil has the greatest eucalyptus production potential, reaching 38.9 m$^3$/ha/year and 7.53 million hectares planted in 2022 [3]. However, physiological disturbances such as attacks by pests, diseases, and weeds can cause losses, resulting in negative impacts on productivity [4,5].

Grasses have a high ability to absorb and use nutrients [6], leading to nutritional imbalance, especially in the initial development of eucalyptus [7]. Therefore, chemical control is widely used in eucalyptus cultivation [8], and about 207 herbicides are registered in Brazil [9]. Among these products, the active ingredients glyphosate, glufosinate ammonium, haloxyfop-p-butyl, clethodim, clomazone, flumioxazin, diuron, indaziflam, sulfentrazone, and *S*-metolachlor stand out [9–13].

Glyphosate is used in agricultural and forestry crops, and the increase in the use of this product was due to the adoption of the technology of genetically modified organisms tolerant to the herbicide [14], which increased the number of resistant weeds [15]. This scenario required control alternatives, such as using herbicides applied in pre-emergence

and mixtures of products with different mechanisms of action [16]. The main crops tolerant to glyphosate are food or used in the production of biofuels [17]. However, transgenic eucalyptus has already been released for planting in Brazil, which can cause changes in the chemical management of weeds in forest stands [18].

*S*-metolachlor is a herbicide derived from chloroacetamide applied pre-emergence or incorporated pre-planting [19]. The product originated from the mixture of *S*-metolachlor and glyphosate, which has systemic action and, as it contains *S*-metolachlor in its formulation, it also has a residual activity on the germination and emergence of weeds [9]. Herbicides applied pre-emergence promote early weed control and help manage resistant biotypes [20–22]. However, the efficiency of these molecules depends on their bioavailability in the solution, the physicochemical characteristics of the soil, the method of preparing the area, and climatic factors during and after application [23–27].

Straw can be found in forest areas where crop residues remain covering the soil, affecting herbicides' effectiveness in subsequent cycles [10]. This is because herbicides, mainly pre-emergent, have to cross this barrier and reach the soil to control the seed bank [28,29]. However, straw also reduces exposure to light, the performance of photosynthesis, and the emergence of weeds [30,31], in addition to benefiting the soil with moisture, nutrients, and an environment favorable for the microbiota [32].

Rain is essential for the herbicide to pass through the straw and reach the soil solution [33], but in dry conditions, these products are retained and adsorbed to the straw, which reduces weed control [34,35]. Thus, the efficiency of pre-emergent herbicides applied under straw depends on the product's characteristics, the nature and amount of straw in the soil, and the occurrence of rainfall [10]. Studying the interaction between these three factors is essential, especially in the forest sector, where these studies are little explored. This work aimed to evaluate the efficiency of the herbicide *S*-metolachlor + glyphosate (Sequence®) in the control of grasses in different densities of eucalyptus straw and with simulated rainfall after application of the product.

## 2. Material and Methods

### 2.1. Location and Soil

The experiment was carried out at the JK Campus of the Federal University of the Vales of Jequitinhonha and Mucuri in Diamantina, Minas Gerais, Brazil, between November and December 2022, in a climate-controlled greenhouse with a minimum temperature of 15.3 °C, a maximum of 43.6 °C, and average 26.8 °C.

The soil was collected in Curvelo, Minas Gerais, Brazil (Table 1), sieved through a 4 mm mesh, and fertilized as recommended for pastures in Minas Gerais, Brazil [36].

**Table 1.** Physical-chemical characteristics of the soil samples used in the experiment.

| pH (H₂O) | P | K | Ca | Mg | Al | H + Al | SB | (t) | T | V | m | MO Dag kg⁻¹ |
|---|---|---|---|---|---|---|---|---|---|---|---|---|
| | mg dm⁻³ | | ----------------Cmolc dm⁻³------------------ | | | | | | | ---%--- | | |
| 5.5 | 2.3 | 88 | 1.24 | 0.44 | 0.1 | 2.97 | 1.91 | 2.01 | 4.88 | 39.1 | 5.0 | 1.9 |
| P-rem | Zn | Fe | Mn | Cu | B | | Sand | | | | | 6 |
| mg L⁻¹ | ----------mg dm⁻³---------- | | | | | | Clay | | | % | | 69 |
| 20.1 | 2.0 | 37.8 | 148. | 0.8 | 0.1 | | Silt | | | | | 25 |

Source: Viçosa Soil Analysis Laboratory, LTDA. pH in water, KCl and CaCl Ratio 1:2.5; P—K—Fe—Zn—Mn—Cu—B—Mehlich extractor 1; Ca—Mg—Al—Extractor: KCl—1 mol/L; H + Al—Calcium Acetate Extractor 0.5 mol/L—pH 7.0; S—Extractor—Monocalcium phosphate in acetic acid; SB—Sum of Exchangeable Bases; CTC (t)—Effective Cation Exchange Capacity; CTC (T)—Cation Exchange Capacity at pH 7.0; V—Base Saturation Index; m—Aluminum Saturation Index; Mat. Org. (MO)—C.Org × 1.724—Walkley–Black; P-rem—Remaining Phosphorus.

### 2.2. Experimental Design

The experiment was designed in randomized blocks, in a 3 × 3 × 2 factorial, with four replications. The first factor represented the doses of the herbicide *S*-metolachlor + glyphosate

(Sequence$^{®}$): 0; 1.06 + 0.79 and 2.12 + 1.59 kg a.i. ha$^{-1}$. The herbicide doses were equivalent to 0, 50, and 100% of the recommended commercial dose for eucalyptus. The second factor represented the straw density: 0, 5, and 10 tons ha$^{-1}$; and the third, the applied water depth: 25 and 50 mm. In total, we obtained 18 treatments and 72 experimental units.

## 2.3. Conducting the Experiment

The grass mix is composed of *Digitaria insularis*, *Urochloa brizantha* cv. marandu, *U. brizantha* cv. piatã and *U. decumbens*, in equal proportions, were sown in trays with an individual capacity of 10 L and an area of 0.28 m$^2$, containing sieved and fertilized soil. Grasses were chosen based on their occurrence in forest plantations and competition with eucalyptus plants. The eucalyptus straw was collected, weighed, and distributed in trays according to the pre-established treatments (0 and 10 tons ha$^{-1}$).

About 24 h after sowing the grasses and adding the straw, the herbicide was applied to the experimental units, according to the defined doses, and with the aid of the electric backpack sprayer Yamaho FT5$^{®}$, the capacity of 5 L.

The rainfall simulation was carried out 24 h after herbicide application, using the sprinklers coupled in the greenhouse, with an intensity of 0.3 mm min$^{-1}$. The blades applied corresponded to 25 and 50 mm, varying according to the treatments. Irrigation was performed daily, maintaining the humidity between 60 and 70% of field capacity.

## 2.4. Evaluated Parameters

The aerial part of the grasses was collected 43 days after the sowing (DAS) and weighed on a precision scale to establish the fresh mass. Subsequently, they were packed in paper bags, taken to an oven with forced air circulation at 60 °C for approximately 96 h until constant mass, and weighed to determine the dry mass.

Visual assessments of the percentage of control were performed at 43 DAS, with scales ranging from 0 to 100, where 0 represents no control and 100 efficient control.

## 2.5. Statistical Analysis

Analysis of variance (ANOVA) was performed using the F-test, and when significant, the means were compared using the Tukey test at 95% probability with the statistical program R$^{®}$ version 4.1.1. Response surface graphs were generated using SigmaPlot$^{®}$ software version 12.0.

## 3. Results and Discussions

The fresh and dry mass of the grasses, in the treatments without the application of *S*-metolachlor + glyphosate and under the effect of the dose of 1.06 + 0.79 kg i.a. ha$^{-1}$, were lower in the density of 10 tons ha$^{-1}$ of straw of eucalyptus, for the dose of 2.12 + 1.59 kg i.a. ha$^{-1}$ there was no significant difference (Table 2). This occurred because the greater volume of straw reduces average daily soil temperature [37] and exposure to light, which are the main regulators of seed germination [38–41]. The unavailability of light also causes the emergence of abnormal grass seedlings, with the development of only the aerial or root part, which limits photosynthesis and nutrient absorption, leading to the death of the seedlings [42]. Grass seeds germinate at a maximum depth of 5.2 and 9 cm [43,44], so the thicker the straw layer, the lower the germination, reducing fresh and dry mass and controlling weeds [45].

In general, the fresh and dry mass and the dry mass/water ratio of the grasses decreased with increasing herbicide dosage (Table 2). The lower fresh and dry mass of grasses was due to *S*-metolachlor, which acts as an inhibitor of very long-chain fatty acids, targeting the condensing enzyme 3-ketoacyl-CoA synthase [46,47]. Very long-chain fatty acids play crucial roles as precursors of triacylglycerols, sphingolipids, and phospholipids used to form waxes, suberin, and cutin [48]. These substances are essential for biological processes such as cell division and differentiation, gibberellin synthesis, and starch conversion into glucose [49]. By reducing fatty acid levels, *S*-metolachlor inhibits growth and causes embryonic mortality [47,50]. In grasses, the product acts on the epicotyl, which generates

reductions in the aerial part and biomass accumulation [32]. In addition, grasses subjected to the herbicide produced less biomass using the same amount of water, which suggests that water use efficiency decreased as a function of the amount of herbicide applied [51,52].

**Table 2.** Fresh mass (g), dry mass (g), and fresh mass/dry mass ratio (g) of grasses at 43 days after sowing under eucalyptus straw and application of *S*-metolachlor + glyphosate.

| Variable | Straw (ton ha$^{-1}$) | *S*-Metolachlor + Glyphosate (kg i.a. ha$^{-1}$) | | | CV% |
| | | 0 | 1.06 + 0.79 | 2.12 + 1.59 | |
|---|---|---|---|---|---|
| Fresh mass (g) | 0 | 155.31 Aa | 54.79 Ab | 12.34 [NS] c | |
| | 5 | 127.33 Aa | 44.08 Ab | 12.92 [NS] c | 21.1 |
| | 10 | 82.71 Ba | 27.13 Bb | 21.63 [NS] c | |
| Dry mass (g) | 0 | 43.10 Aa | 8.92 Ab | 1.32 Cc | |
| | 5 | 32.86 Ba | 7.82 Ab | 1.73 ABc | 26.1 |
| | 10 | 19.15 Ca | 3.90 Bb | 2.92 Ab | |
| Dry mass/water ratio | 0 | 0.397 Aa | 0.216 [NS] b | 0.105 [NS] c | |
| | 5 | 0.348 ABa | 0.217 [NS] b | 0.156 [NS] c | 20.3 |
| | 10 | 0.299 Ba | 0.175 [NS] b | 0.150 [NS] b | |

NS—not significant by F-test at 95% probability. Means followed by the same letter, case in the column, and lowercase in the row do not differ by the Tukey test at 95% probability. CV—Coefficient of variation (%).

Treatments without straw and with the herbicide, regardless of the water level applied, obtained the highest percentages of control, the highest at the dosage of 2.12 + 1.59 kg i.a. ha$^{-1}$ of *S*-metolachlor + glyphosate. (Figure 1). The greater efficiency in the isolated use of *S*-metolachlor + glyphosate occurred because straw treatments favored the herbicide's adsorption and biodegradation. Although glyphosate and *S*-metolachlor generally have low adsorption [53–55], soil cover from forest species generally has a higher presence of lignin and cellulose, which generates greater adsorptive capacity [56]. Furthermore, the main degradation route of *S*-metolachlor is microbial [57]. The biodegradation of the herbicide is affected by the straw's type and state of degradation. Still, it is known that the presence of cover on the soil increases the supply of carbon, which favors microbial activity, accelerating the dissipation of *S*-metolachlor [53,58]. In general, the herbicide straw interaction controlled the grasses; however, the greater efficiency in the isolated use of *S*-metolachlor + glyphosate suggests that the presence of vegetation cover may have an antagonistic effect on the product.

There was no significant difference in the percentage of grass control depending on the rainfall simulation (25 or 50 mm) (Figure 2). The water depth applied did not influence the control of grasses because 20 to 50 mm of precipitation is considered ideal for pre-emergent herbicides to cross the residues present in agricultural and forest areas and reach the soil [10,59,60]. Harvest residues can retain, degrade, and intercept herbicides [10]; therefore, the occurrence and volume of rain after application can influence the efficiency of chemical control [61]. In this case, 25 and 50 mm precipitations facilitated the herbicide molecule to cross the straw and reach the soil, controlling the grasses.

*S*-metolachlor + glyphosate (kg i.a. ha⁻¹)

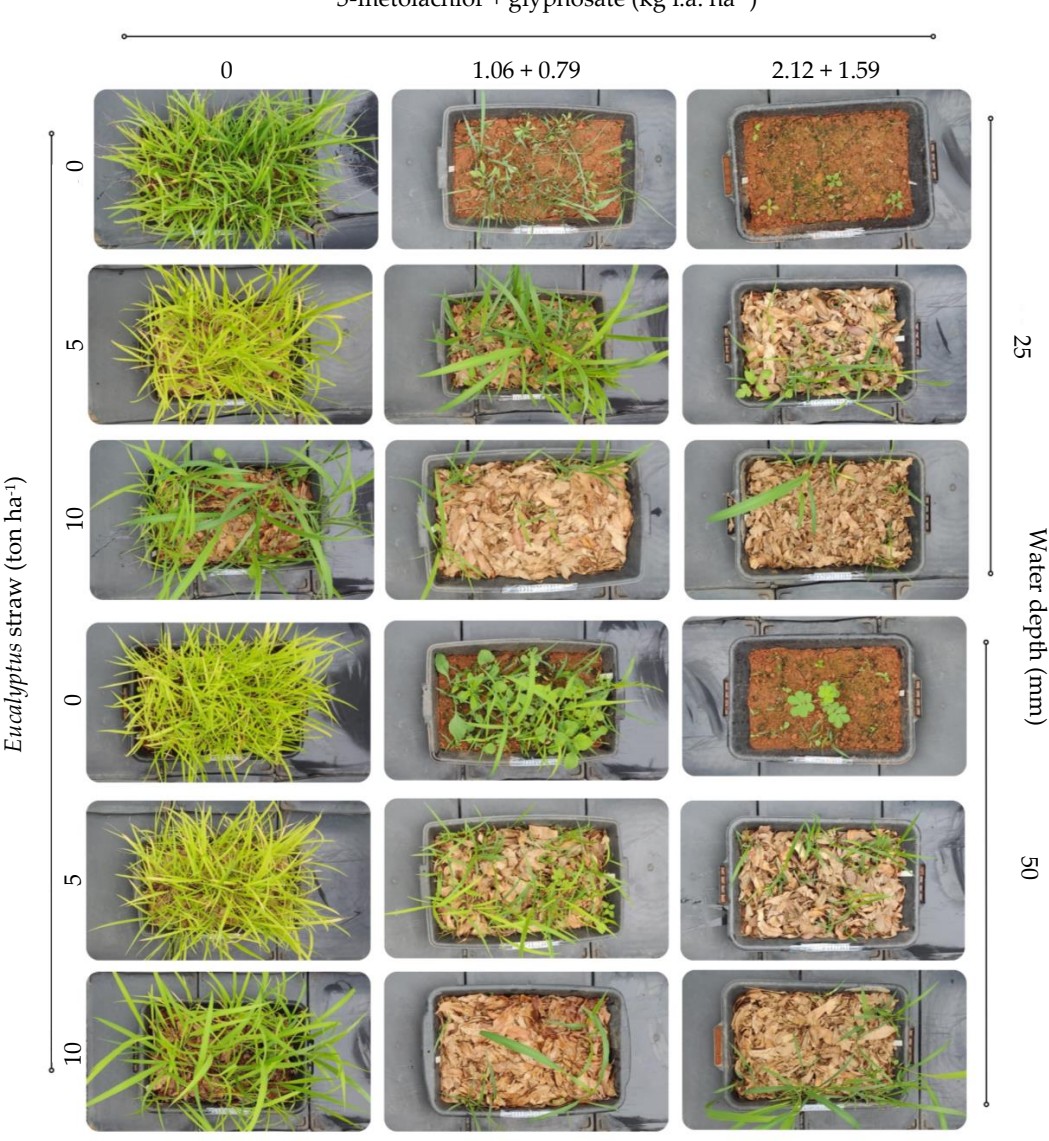

**Figure 1.** Grasses 43 days after sowing under eucalyptus straw, application of *S*-metolachlor + glyphosate, and water slide.

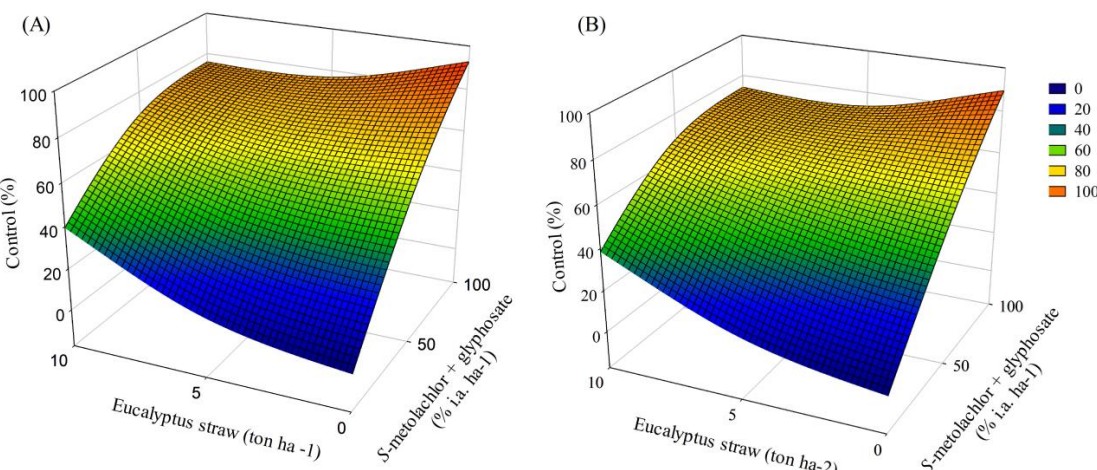

**Figure 2.** Surface response of the percentage of grass control at 43 days after sowing, under eucalyptus straw, application of *S*-metolachlor + glyphosate, and water depth of 25 (**A**) and 50 mm (**B**).

## 4. Conclusions

Eucalyptus straw alone (10 tons ha$^{-1}$) exerts approximately 40% control over grasses, reducing fresh and dry mass. Therefore, depending on the level of infestation and the amount of straw in the area, soil cover may be sufficient to promote grass control.

In treatments containing straw and herbicide, *S*-metolachlor + glyphosate was partially adsorbed by the vegetation cover and biodegraded by microorganisms, decreasing the efficiency of the product. This is observed because the greatest control occurred without straw, mainly at the 2.12 + 1.59 kg i.a. ha$^{-1}$ dosage. Thus, we can conclude that the vegetation cover can antagonize the herbicide due to the high concentration of lignin and cellulose in the straw and the favoring of microbial activity, accelerating the degradation of *S*-metolachlor + glyphosate.

In future experiments, it would be interesting to test other levels of water depth since those applied in this study (25 and 50 mm) did not influence the fresh, dry, and control weight of the grasses. In addition, new studies are needed that discuss the interaction between herbicide, water depth, and straw in the forest sector, studying the effect of other pre-emergent herbicides, broadleaf weed control, and the residual potential of these products.

**Author Contributions:** Conceptualization, T.S.D. and F.S.O.; Methodology, T.S.D. and J.B.d.S.; Software, F.S.O. and I.M.S.; Validation, B.C.C.F., L.L.L.d.S.R. and D.V.S.; Formal Analysis, T.S.D. and F.S.O.; Investigation, T.S.D. and F.S.O.; Resources, I.M.S. and B.C.C.F.; Writing—Original Draft Preparation, T.S.D. and F.S.O.; Writing—Review and Editing, J.B.d.S. and D.V.S.; Supervision, J.B.d.S. and D.V.S. All authors have read and agreed to the published version of the manuscript.

**Funding:** This work is supported by Conselho Nacional de Desenvolvimento Científico e Tecnológico (CNPq), Coordenação de Aperfeiçoamento de Pessoal de Nível Superior (CAPES)—Código Financeiro 001 and Fundação de Amparo à Pesquisa do Estado de Minas Gerais (FAPEMIG).

**Data Availability Statement:** Data are contained within the article.

**Acknowledgments:** The authors wish to thank Conselho Nacional de Desenvolvimento Científico e Tecnológico (CNPq), Coordenação de Aperfeiçoamento de Pessoal de Nível Superior (CAPES)—Código Financeiro 001, Fundação de Amparo à Pesquisa do Estado de Minas Gerais (FAPEMIG), Programa MAI DAI CNPq/ Syngenta/ UFVJM.

**Conflicts of Interest:** The authors declare that they have no conflict of interest.

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
