# Peer review of "Efficacy of S-metolachlor + glyphosate for Weed Control in Different Levels of Eucalyptus Straw"

_forests, doi:10.3390/f14091828_

Round 1
Reviewer 1 Report
The manuscript entitled “Efficacy of S-metolachlor + glyphosate for weed control in different levels of eucalyptus straw” has a fascinating topic, well written and structured work. However, some critical revisions are required. The following revisions must be considered:
General comments:
Abstracts focus primarily on materials and methods. In your abstract, you may provide more statistical results and a general conclusion.
A randomized block was used in a factorial arrangement as described in your abstract line 26. How is DBC abbreviated?
- Introduction, lines 48-49: Indicate the year for this statement “Brazil has the greatest eucalyptus production potential, with increasing productivity, reaching 38.9 m³/ha/year and 7.53 million hectares planted”.
- Introduction, lines 60-61: Provide registered pesticides and herbicides for pest and weed control in eucalyptus in Brazil.
- Material and methods, Table 1: Describe the carbon content.
- Material and methods, lines 142-143: You missed 5 tons ha-2 for pre-established treatments.
- Material and methods, lines 154-155: You stated “The aerial part of the grasses was collected 43 days after the sowing of the grasses”. What are the reasons for 43 days? What was the status of weeds in 43 days (Vegetative stage)?
- Material and methods, lines 156-158: Why did you not consider the roots, since root dry weight is an important factor in such studies?
- Material and methods, line 157: Mention the time of oven drying process.
- Material and methods, line 165: Which package of R software was used in this experiment? (drc or what?)
Results and discussion, lines 169-180: What about the herbicide dose of 2.12+1.59 kg ha-1?
- Table 2, no need to provide all details in the title.
- Table 2: Did you calculate the biomass of all grasses at the same time? Weeds weren't considered separately, why?
- Table 2: Increasing straw volume reduced the fresh and dry mass of grasses, according to your findings. Nevertheless, when 10 tons of straw were applied per ha, the biomass was increased. Is there a reason for this, in your opinion?
- Figure 1: Provide a higher quality version.
- Figure 1: What are the broad leaf plants in the fourth row (as you sow the grass)?
- Results and discussion, lines 233-236: You stated that “Therefore, the occurrence and volume of rain after application can influence the efficiency of chemical control [56]; mainly causing the molecule to reach the soil and not remain in the straw, where microbial degradation may occur”. You mean that microbial degradation didn't occur in the soil environment, where microorganisms live there?!
- Figure 2: These two charts appear to be similar. Visually, neither chart differs from the other. Explain that.
There is a lack of quality in the discussion section, and it is not acceptable for a research paper. It is not clear to me what the requirements and standards are for a "Brief Report".
- Conclusion: Results are just repeated!
It is necessary to moderately edit the text since it contains some vague sentences.
Author Response
Dear Reviewer 1,
Comment: The manuscript entitled “Efficacy of S-metolachlor + glyphosate for weed control in different levels of eucalyptus straw” has a fascinating topic, well written and structured work. However, some critical revisions are required. The following revisions must be considered:
Response: Thanks for the comment. Thank you for reviewing and for your interest in our article. All suggestions will be corrected by the authors. The essay in English was revised by a professor in the area. If necessary, we are willing to make further revisions. We've made changes to all topics and increased the number of references we feel are necessary. All changes are highlighted in red in the manuscript.
Comment 1: Abstracts focus primarily on materials and methods. In your abstract, you may provide more statistical results and a general conclusion.
Response 1: Done! In the abstract, we reduced the information regarding the methodology and inserted more information on results and conclusions.
Comment 2: A randomized block was used in a factorial arrangement as described in your abstract line 26. How is DBC abbreviated?
Response 2: Done! In the abstract we write in full the statistical design used.
Comment 3: - Introduction, lines 48-49: Indicate the year for this statement “Brazil has the greatest eucalyptus production potential, with increasing productivity, reaching 38.9 m³/ha/year and 7.53 million hectares planted”.
Response 3: Done, the information has been added in the text! The data mentioned are from the Brazilian Tree Industry (IBÁ), for the year 2022.
Comment 4: - Introduction, lines 60-61: Provide registered pesticides and herbicides for pest and weed control in eucalyptus in Brazil.
Response 4: Done! We have inserted in the text some examples of herbicides, which are registered for eucalyptus cultivation in Brazil. Due to the large volume of registered products, it is not possible to list them all, but we have included the most used ones. Because our work is focused on the use of herbicides, we chose to mention only molecules of this class of pesticides.
Comment 5 - Material and methods, Table 1: Describe the carbon content.
Response 5: Table 1 shows all the physical and chemical analysis available for the soil used in the experiment. The quantification and extraction methods are described in the table caption.
Comment 6: - Material and methods, lines 142-143: You missed 5 tons ha-2 for pre-established treatments.
Response 6: Yes, our treatments for straw in the soil are 0, 5 and 10 ton ha-2. We defined this amount according to the material available (trays and straw) and because our objective is to observe differences in weed control based on the superficial thickness of the straw on the soil. With the straw levels applied, such a difference could be more evident.
Comment 7: - Material and methods, lines 154-155: You stated “The aerial part of the grasses was collected 43 days after the sowing of the grasses”. What are the reasons for 43 days? What was the status of weeds in 43 days (Vegetative stage)?
Response 7: We opted for a duration of 43 days because we wanted to observe the efficiency of straw + herbicide control mainly during the initial period of eucalyptus development. Since it is known that the greatest damage attributed to weeds in eucalyptus cultivation is in the first days after transplanting (Maciel et al., 2020). Furthermore, as we are evaluating a herbicide with a residual effect, S metolachlor, we chose the duration based on the half-life of the product which varies between 6 and 100 days (Westra et al., 2020).
Our experiment had a lack of uniformity in the emergence of grasses according to the treatments, due to the action of the herbicide and the straw. In addition to using a mix of seeds, not just one species. Therefore, we cannot state the vegetative stage of all experimental units at the end of 43 days, but we know that in general the grasses were between stages V3 and V6.
Maciel, J. C.; Duque, T. S.; Ferreira, E. A.; Zanuncio, J. C.; Plata-Rueda, A.; Silva, V. P.; Dos Santos, J. B. Growth, Nutrient Accumulation, and Nutritional Efficiency of a Clonal Eucalyptus Hybrid in Competition with Grasses. Forests, 2022, 13, 1157. http://dx.doi.org/10.3390/f13081157.
Westra, E. P. et al. Evaluation of sorption coefficients for pyroxasulfone, S-metolachlor, and dimethenamid-p. Air, Soiland Water Research, 2015, 8.
Comment 8: - Material and methods, lines 156-158: Why did you not consider the roots, since root dry weight is an important factor in such studies?
Response 8: In our experiment we considered only the aerial part to mainly observe the interference of the straw layer in the emergence of weeds. Furthermore, in grasses, the main site of action of S-metolachlor is in the aerial part, interrupting cell division and multiplication. However, I agree that collecting roots would add to our work, thanks for the suggestion, we will apply it in future work.
Comment 9: - Material and methods, line 157: Mention the time of oven drying process.
Response 9: Done! Insert in the text!
Comment 10: - Material and methods, line 165: Which package of R software was used in this experiment? (drc or what?)
Response 10: During statistical analysis in R, we used various packages for analysis of assumptions (independence of errors, normality, uniformity of variances and additivity), ANOVA and Tukey Test. But some of the packages used were: lmtest, car, dae, asbio, agricolae and ExpDest.pt.
Comment 11: Results and discussion, lines 169-180: What about the herbicide dose of 2.12+1.59 kg ha-1?
Response 11: Done! We insert the information in the text.
Comment 12: - Table 2, no need to provide all details in the title.
Response 12: We reduced the information in the captions of figures and tables. However, we always opt for detailed captions so that, if the reader only sees the table, he understands all the data present.
Comment 13: - Table 2: Did you calculate the biomass of all grasses at the same time? Weeds weren't considered separately, why?
Response 13: Yes, we do not calculate biomass separately. That's because we wanted to simulate what happens in the field. In Brazilian forest areas, different species of grasses occur simultaneously. Our objective was to observe the general control of this grass population. In addition, according to the package leaflet, the herbicide is effective in controlling grasses. But could it be that when applied on straw and depending on the depth of the water, the efficiency was maintained? These are our hypotheses.
Comment 14: - Table 2: Increasing straw volume reduced the fresh and dry mass of grasses, according to your findings. Nevertheless, when 10 tons of straw were applied per ha, the biomass was increased. Is there a reason for this, in your opinion?
Response 14: In table 2, we observe that this pattern of biomass increase in 10 tons of straw per ha only occurs at the dosage of 2.12 + 1.59 kg i.a. ha-1. In my opinion, with a higher dosage of hebicide and straw, there is adsorption and physical impediment of the product by the straw, reducing control, resulting in a greater number of grasses and consequently biomass.
So, we see that the straw without the herbicide provided control of the grasses, but that there is a straw - herbicide interaction and depending on the amount of both, it becomes antagonistic. Therefore, the greatest control observed was at the dose of 2.12 + 1.59 kg i.a. ha-1 without straw. This remark can be found on lines 208-212.
Comment 15: - Figure 1: Provide a higher quality version.
Response 15: Done! We insert the highest quality image.
Comment 16: - Figure 1: What are the broad leaf plants in the fourth row (as you sow the grass)?
Response 16: During our experiment, we observed that some broadleaf weeds emerged. We believe it occurred because of seeds contained in the soil. However, in the control evaluations, we disregarded broadleaf weeds, as we were targeting grasses. Some observed species: Senna obtusifolia, Bidens pilosa, Richardia brasiliensis, Sonchus oleraceus and Emilia fosbergii.
Comment 17: - Results and discussion, lines 233-236: You stated that “Therefore, the occurrence and volume of rain after application can influence the efficiency of chemical control [56]; mainly causing the molecule to reach the soil and not remain in the straw, where microbial degradation may occur”. You mean that microbial degradation didn't occur in the soil environment, where microorganisms live there?!
Response 17: I understand your question, we recognize that the herbicide can also be degraded by the soil microbiota. The sentence was confusing, so let's rewrite it, so that it is better understood. We have inserted more discussion into the work on this.
Comment 18: - Figure 2: These two charts appear to be similar. Visually, neither chart differs from the other. Explain that.
Response 18: Of course. The graphs refer to the control of grasses as a function of the applied water depth (25 or 50mm). For this variable, we did not obtain a significant result (lines 216 and 217), however, we chose to keep the graphs because they bring relevant information about the control. For example, from the graph we observe that in the treatment of 10 ton of straw without herbicide, 40% of grass control already occurs. And, in the treatment without straw and with 2.12 + 1.59 kg i.a. ha-1 of S-metolachlor + glyphosate, the control approaches 100%.
Comment 19: There is a lack of quality in the discussion section, and it is not acceptable for a research paper. It is not clear to me what the requirements and standards are for a "Brief Report".
Response 19: We rewrote part of the work discussions, and inserted discussions related to physiological characteristics. Changes are highlighted in red.
Comment 20: Conclusion: Results are just repeated!
Response 20: Done! The conclusions were rewritten. Furthermore, we have added a graphical abstract to illustrate our experimental design and the results of this research.
Reviewer 2 Report
The article examines a current problem regarding the possibilities of weed control by applying herbicides and mulching with straw as a non-chemical method of control. The obtained results allow a choice to be made, depending on the degree of weeding, which of the two approaches to apply. The researches were carried out under a correct methodological setting. The article submitted for review contains the necessary sections – introduction, material and methods, results and conclusion. It is formatted as required.
Author Response
Dear Reviewer 2,
Comment: The article examines a current problem regarding the possibilities of weed control by applying herbicides and mulching with straw as a non-chemical method of control. The obtained results allow a choice to be made, depending on the degree of weeding, which of the two approaches to apply. The researches were carried out under a correct methodological setting. The article submitted for review contains the necessary sections – introduction, material and methods, results and conclusion. It is formatted as required.
Response: Thanks for the comment. Thank you for reviewing and for your interest in our article. All suggestions will be corrected by the authors. The essay in English was revised by a professor in the area. If necessary, we are willing to make further revisions. We've made changes to all topics and increased the number of references we feel are necessary. All changes are highlighted in red in the manuscript.
Reviewer 3 Report
The brief report depicts that the application of 2.12+1.59 kg/ha of S-metolachlor + glyphosate demonstrates the best result over weed control. The report needs extensive revisions as follows:
Why did the authors take these two chemicals?
There is no need to mention the a.i. since S-metolachlor and glyphosate means 100% chemical a.i.
What are the research gaps?
Treatment details are not clear. Need a clear depiction.
The discussion part is poorly written. The author must explore the physiological aspects of weed control in this study.
Similarly, the conclusion part must be rewritten as per the objective of the study.
Author Response
Dear Reviewer 3,
Thanks for the comment. Thank you for reviewing and for your interest in our article. All suggestions will be corrected by the authors. The essay in English was revised by a professor in the area. If necessary, we are willing to make further revisions. We've made changes to all topics and increased the number of references we feel are necessary. All changes are highlighted in red in the manuscript.
Comment 1: The brief report depicts that the application of 2.12+1.59 kg/ha of S-metolachlor + glyphosate demonstrates the best result over weed control. The report needs extensive revisions as follows: Why did the authors take these two chemicals?
Response 1: It is known that numerous glyphosate-resistant weed biotypes, including grasses, have emerged in recent decades. Therefore, as an alternative, mixtures of herbicide active ingredients have been launched on the market. Therefore, the first factor for us to choose this product was to test the effectiveness of the herbicide that encompasses the pre-emergent effect of S-metolachlor with the post-emergent effect of glyphosate. The second factor was the scarcity of works related to the interference of straw in chemical control in forest plantations. The third factor is because this is the only herbicide containing S-metolachlor registered for eucalyptus plantations in Brazil, a product launched recently, therefore, research is needed that encompasses the effect of this herbicide with different management and environmental conditions.
Comment 2: There is no need to mention the a.i. since S-metolachlor and glyphosate means 100% chemical a.i.
Response 2: During the text, we chose to standardize the value of kg i.a. ha-1 because this dosage used refers to that recommended for eucalyptus. This standardization allows the work to be recreated using different brands of registered products, but with the same active ingredient. If we chose to use, for example, only the nomenclature 100% i.a. ha-1, this percentage would vary according to the crop in question.
Comment 3: What are the research gaps?
Response 3: Our work is an initial experiment that gives a good answer to the interaction between herbicide, water depth and straw in the forestry sector. However, as it is one of the pioneering works in this scheme for the forest sector, there are some gaps for future work, such as studying the effect of this herbicide on broadleaf weeds. Conduct a long-term experiment to determine the residual effect of S-metolachlor when applied to straw and when further applications should be made. Studies that include the effect of the herbicide on the roots and their collection. And mainly, repeat the experiment with other herbicide molecules. Some of these gaps were added at the conclusion of the manuscript.
Comment 4: Treatment details are not clear. Need a clear depiction.
Response 4: In this work we included three factors: herbicide, straw and water depth. The herbicidal factor has the levels: 0; 1.06 + 0.79 and 2.12 + 1.59 kg a.i. ha-1. Straw factor: 0, 5 and 10 ton ha -2 e; water depth after herbicide application: 25 and 50 mm. Thus, we obtain 18 treatments with 4 repetitions, totaling 72 experimental units. To make it clearer, we've rewritten the "Experimental Design" topic.
Comment 5: The discussion part is poorly written. The author must explore the physiological aspects of weed control in this study.
Response 5: We redid part of the work discussions, focusing on some physiological characteristics, the changes are highlighted in red. However, in the "Brief Report" format there is a character limitation, which prevents us from extending this section too much.
Comment 6: Similarly, the conclusion part must be rewritten as per the objective of the study.
Response 6: Done! The conclusions were rewritten. Furthermore, we have added a graphical abstract to illustrate our experimental design and the results of this research.
Round 2
Reviewer 1 Report
All my concerns have been addressed by the authors.
Reviewer 3 Report
The authors revised the paper well. In my view, it may be accepted in this journal, now.